# Upholding Quality and Patient Safety during COVID-19 Pandemic—A Jordanian Case Study

**DOI:** 10.3390/healthcare11040523

**Published:** 2023-02-10

**Authors:** Eman Zmaily Dahmash, Thaira Madi, Samar Khaled Hassan, Yazan Oroud, Ahmad Shatat, Rawan Jalabneh, Hafez Abu Rashideh, Aws Aljayyousi, Affiong Iyire

**Affiliations:** 1Faculty of Health, Science, Social Care and Education, School of Life Sciences, Pharmacy and Chemistry, Kingston University, London KT1 2EE, UK; 2Department of Accreditation, Healthcare Accreditation Council, Amman 11181, Jordan; 3Department of Accounting, Faculty of Business, Isra University, Amman 11622, Jordan; 4Aston Pharmacy School, College of Health & Life Sciences, Aston University, Birmingham B4 7ET, UK

**Keywords:** COVID-19 pandemic, readiness, hospitals, emergency preparedness, accreditation, patient safety

## Abstract

Background: The advent of the COVID-19 pandemic caused a rapid increase in demand for healthcare services over a prolonged period, and the hospital emergency preparedness system has been essential. Therefore, this study aimed to explore Jordanian hospitals’ response to emergency situations and examine the underlying role and effect of accreditation programs as a “Quality and Patient Safety” tool to deal with emergency situations during the pandemic. Methods: An online survey for a cross-sectional study was conducted in Jordan between 1 March and 30 May 2022, to examine the opinions of hospitals’ top, senior, and middle managers using a validated questionnaire. Results: A total of 200 healthcare providers from 30 hospitals participated in the study. From the areas within accreditation standards that were investigated, capacity building on emergency preparedness and communication abilities received the least scores (2.46 and 2.48, respectively). Additionally, hospitals with mature quality and patient safety culture (>3 accreditation cycles) demonstrated a statistically significant difference in score in two domains–emergency preparedness (*p* = 0.027) and infection prevention and control (*p* = 0.024). Conclusions: During outbreaks, hospitals that are required to comply with accreditation standards that address all emergency preparedness aspects will fare better in quality performance.

## 1. Introduction

It would be difficult for a hospital, even one that is well-prepared, to handle the health consequences of the COVID-19 outbreak. During emergent pandemics, healthcare systems are expected to be significantly impacted by staff shortages and critical equipment/supplies shortages, which can lead to limitations in access to necessary care and can directly influence healthcare delivery. An unprepared healthcare facility’s ability to provide acute treatment might be hampered by the cessation of essential support services and supplies. Therefore, the readiness of healthcare organizations and the implemented systems to deal with such draining and challenging environments are key factors that would determine the outcomes and may enable hospital-based management to become more effective during an escalating epidemic [1,2,3].

The dynamic development of the hospital preparedness and response system was essential to combat the outcomes of COVID-19, slow further distribution of the infection, and prevent healthcare systems from becoming overwhelmed due to critically ill patients infected with COVID-19 [4,5,6]. A key consideration is the hospital’s resilience and ability to adapt and grow beyond what is normally possible to provide more clinical treatment, as COVID-19 instances may result in an extended period of rapidly rising demand. Therefore, a hospital is required to take proactive measures to estimate care demands, identify gaps in critical care, and identify maximal case admission capacities [7]. A hospital’s emergency response system to the COVID-19 pandemic must consist of multiple interconnected components, such as: The emergency management plan, infection prevention and control program, staff capacity building, clinical case management, communication coordination, and laboratory and diagnostic services [1,8,9].

Quality and patient safety standards are fundamental in ensuring the proper team involvement, the adoption of evidence-based practices and policies, accessibility to essential medical equipment, diagnostic procedures, and personal protective equipment, the provision of safe care, and proper channels for communication and coordination [10,11]. Accreditation processes can result in multifaceted improvements in the standard of treatment. These standards require establishing and adhering to clear administrative and clinical practices that establish clear standards and encourage information exchange and staff and/or patient issue resolution. Studies have shown that such standards support the development of a “culture of quality” and motivate personnel to perform to the best of their abilities [12,13,14].

The accreditation process encourages behaviors that formalize and build platforms for patients’ and staff members’ views to be heard. Additionally, accreditation criteria encourage staff members to speak out and participate in committee discussions to address issues including infection control, quality improvement, patient safety, and security. Staff meetings are organized on a regular basis to facilitate this. Accreditation-mandated transparency fosters and sustains a culture of respectful patient-centered care [15].

To enhance staff and patient safety during emergencies, national and international organizations have developed various guiding documents in different formats, such as guidelines, frameworks, or standards. These documents are developed for international use (e.g., the World Health Organization (WHO) [5,16] and the Centre of Disease Control and Prevention (CDC) [4,8]), or for external evaluation purposes, like the Center for Medicare and Medicaid services (CMS) [17,18], National Health Services—UK (NHS) [19], Occupational Safety and Health Administration (OSHA) [20], the Joint Commission [21], Accreditation Canada [22], Healthcare Accreditation Council (HCAC) [9], and the ISO [23]. They are usually linked with the national regulations and requirements.

Accreditation programs, worldwide, aim to foster quality improvement and patient safety and assure that the healthcare institutions have a system for complying with standards. Most of these standards are accredited by the International Society for Quality in Healthcare External Evaluation Association (IEEA) [10] and meet IEEA requirements. Staff ongoing education, staff wellbeing promotion, use of evidence-based practices, internal and external coordination, risk management, staff and patient safety, etc., are part of the IEEA requirements. Accordingly, emergency preparedness, in general, is a core element of the accreditation standards [5,10,12,14].

Few studies have been carried out to assess the overall impact of accreditation. A systematic review conducted by Alkhenizan and Shaw in 2011 [12] revealed evidence that accreditation enhanced the care process offered by healthcare services and improved clinical results. Another study by Hussein et al. [13] reported that accreditation stimulated performance improvement and patient safety. Moreover, a study in Uganda to assess the influence of laboratory accreditation on the performance of healthcare services reflected that there was a positive impact of accreditation on increasing healthcare services performance in accredited facilities [14].

During the COVID-19 pandemic, accreditation organizations played an integral role in supporting healthcare institutions by providing them with needed resources and tools, training sessions, webinars, as well as assessing their readiness to deal with the pandemic. They also reviewed their standards to guarantee addressing the pandemic-related requirements [1,7].

Generally, the accreditation standards include different functional areas worldwide, and emergency preparedness and pandemic management are part of these functional areas. For example, in Jordan, the hospital accreditation standards [24] and the emergency preparedness requirements are integrated in eleven out of fifteen clusters (functional areas), including access and continuity of care, patient care, diagnostic services, infection prevention and control, support services, environmental health and safety, quality improvement and patient safety, information management, human resources management, governance and leadership, and education and training. These requirements enabled healthcare institutions to respond to emergencies and disasters in general and COVID-19 in particular [24]. One study conducted in Turkey revealed that the Turkish Healthcare Quality and Accreditation Institute standards were 79.6% related to the COVID-19 response in general [11].

Another study was conducted to identify the alignment between the Public Health Accreditation Board (PHAB) standards and the Centers for Disease Control and Prevention’s (CDC’s) public health preparedness (PHP). The study’s findings underlined the combined effort between the infrastructural and fundamental aspects represented by accreditation and specific programming actions supported by preparedness financing, with links established in 11 of the 12 PHAB Domains [25]. From these studies, we can infer that accreditation could be used as a monitoring tool to ensure the institution’s preparedness.

As the COVID-19 pandemic is over, this is the right time to assess the performance of healthcare organizations during the pandemic and identify gaps in emergency preparedness processes. This will enable healthcare facilities, policymakers, and accrediting organizations to embrace a change that will promote better preparedness for the next pandemic. Therefore, this research was designed to analyze Jordanian hospitals response to emergency situations and examine the underlying role and effect of the accreditation program as a “Quality and Patient Safety” tool on hospitals’ readiness to deal with emergency situations during the COVID-19 pandemic.

## 2. Materials and Methods

### 2.1. Study Design and Study Population

A cross-sectional study by means of an online survey was conducted in Jordan between 1 March 2022 and 30 May 2022 within medium and large hospital settings across the different health sectors in Jordan. The primary source of data was derived from questionnaires filled by members of emergency response teams that were formed to respond to the COVID-19 pandemic. The inclusion criteria comprised medium and large hospitals, accredited and non-accredited hospitals, within the private and public healthcare sectors. Small-sized hospitals were excluded from the study since the efforts of dealing with COVID-19 were centralized and delegated to medium and large referral hospitals. Primary healthcare centers and ambulatory clinics were also excluded.

A questionnaire was used as the research tool. It was distributed to a minimum of 200 participants (hospital managers, assistant hospital managers, heads of departments and units, heads of infection prevention and control, quality directors, and support services managers), from within the selected hospitals according to prior analysis that was executed during the preparation phase of the project.

### 2.2. Sampling Strategy

A convenient sample of 35 eligible medium and large hospitals was invited via official letters to participate in the study, from which 30 hospitals agreed to participate. An investigator was assigned to contact the management in each hospital through a contact officer that was assigned by the management. The investigator sent the link of the study to the contact officer in each hospital, whose responsibility was to distribute the link to key top senior and middle management staff. All participants voluntarily participated in the study and were thus considered exempt from written informed consent which was included at the beginning of the tool before starting. At the start of the survey questionnaire, the purpose and goals of the study were made clear. The timing of the study was intended to enable hospital managers and leaders to reflect on their practice after the peak of the pandemic, and to enable them to assess the emergency preparedness practices after the pandemic slowed down.

### 2.3. The Questionnaire

Three components made up the questionnaire. The initial segment of the study concentrated on the participants’ backgrounds and demographic data (age, gender, educational level, and job position). The second segment entailed questions that describe and categorize the characteristics of the hospital (sector, geographic location, accreditation status and accreditation cycle, and scope of service pertinent to COVID-19). With regards to the scope of service, the focus was on whether the hospital was authorized to admit and deal with COVID-19 cases, or if the hospital was authorized to quarantine suspected cases and whether the hospital’s laboratory was authorized to deal with COVID-19 samples, or if the hospital dealt with suspected emergency department (ER) cases and referred them to other hospitals. The third segment dealt with emergency response focus areas. In that section, a three-point Likert scale was employed, and the questions were further divided into six focus areas. Each category addressed key emergency response domains. The selection of the questions was based on a literature review and the previously used WHO tool for the assessment of COVID-19 readiness of hospitals. Within the third category, emergency response focus areas included: Emergency preparedness, infection prevention and control, capacity building, case management, communication, and laboratory services. Section 3 was made up of 23 questions, each question was based on a three-point Likert scale (low—1 mark, medium—2 marks, and high—3 marks). The questions in Section 3 represented important steps in accordance with international criteria for emergency preparedness [1,9]. The questions addressed practices in relation to emergency preparedness during the pandemic. The tool asked the participants about the degree with which they agreed with the impact of each question on their readiness to provide care in the COVID-19 pandemic. The participants’ responses ranged from 1 to 3, and the average number of participants certified as having completed each portion was determined. The cut-off score for good preparedness and hence the quality of care provided during the pandemic was set at 83.5% (which was the average score value for the whole study sample).

### 2.4. Ethical Approval

A consent statement was included at the beginning of the online survey explaining the aim of the research, indicating that participation in this research is entirely voluntary and will have no direct benefits to participants and no bearing on their job or on any work-related evaluations. Therefore, all study participants who proceeded to fill and submit their survey gave their informed consent for inclusion before they participated in the study. The study protocol was approved by the World Health Organization (WHO) 2021/1186281-0 as well as the Research Ethics Committee at Isra University, Jordan (SREC/22/03/032), and the Ministry of Health in Jordan (IRB approval no. 1755144).

### 2.5. Statistical Analysis

SPSS software version 25 was used to analyze the data. The demographic features of the subjects were analyzed using descriptive statistics. For normally distributed variables, data were given as mean ± SD, whereas categorical data was provided as percentages (frequencies). Cronbach’s alpha was employed to measure the reliability of the tool with a set of 0.8 as the minimum scale of reliability. Two-way ANOVA was employed to measure significance with a significance level set at 0.05.

## 3. Results

Cronbach’s alpha for scale reliability for the whole tool (based on the 23 questions) was 0.963 indicating high reliability of the tool. The result indicates that there is no need to drop any of the tool questions, which were used for the analysis.

### 3.1. Background and Demographics

A total of 200 healthcare providers from 30 hospitals representing the top, senior, and middle management, participated in the study. Table 1 details the baseline characteristics of the participants. Most of the participants (*n* = 126, 63%) were females, the largest proportion of the participants was among senior management (*n* = 76, 38%) within the age group of 41–50 years (*n* = 72, 36%), and with positions as medical and nursing managers (*n* = 79, 39.5%). Furthermore, the vast majority of participants held a bachelor’s degree (*n* = 127, 63.5%).

Table 2 presents the distribution of participants according to the demographics of their organizations. Just above half of the participants were working in private hospitals (*n* = 104, 52%) and only 9.5% of the participants were from hospitals within the southern region. Overall, 92.5% (*n* = 185) of participants were from accredited hospitals (both national and or international). Around a third of the participants (33%) represented hospitals that had been granted accreditation for more than three cycles, hence, representing mature organizations.

### 3.2. Responses According to Patient Safety Domains

The assessment tool was based on six patient safety areas. Table 3 summarizes the responses to each question. Overall, the results showed that there were variances by domain. The provision of adequate supplies of personal protective equipment (score 90%) and inventory management system (score 86.4%) were, respectively, the most crucial aspects in the domains of infection prevention and emergency preparedness that contributed to increasing the quality of services offered to combat the pandemic. Only 51% of healthcare managers believed that the available educational materials regarding COVID-19 were simplified and visible to the employees. However, the use of management protocols was perceived by managers as the most valuable within the case management domain. The lowest score in communication domain was the usefulness of public information spokespersons in enhancing the communication (80.5%). Laboratory services played a major role during the pandemic, where specimen collection and transportation protocols contributed positively to preventing the incidence of COVID-19 (66% of participants reported its high importance).

### 3.3. Effect of Hospital Characteristics on Response to the Pandemic

The average score of participants’ opinions, when stratified according to hospital type, showed a statistically significant difference among participants (Table 4), where managers at private hospitals scored higher than those at public hospitals in all domains. Furthermore, for all domains, upholding of quality was significantly higher in hospitals within the middle region of Jordan, followed by the southern region, then the northern. Interestingly, despite a statistically insignificant difference, leaders and managers within accredited hospitals perceived higher effect of patient safety and quality aspects on the management of the pandemic than non-accredited facilities. When hospitals were further stratified according to the number of accreditation cycles, which reflect the presence of mature quality and patient safety culture, the results revealed a statistically significant difference in scores in two domains: Emergency preparedness (*p* = 0.027) and infection prevention and control (*p* = 0.024).

To further understand the correlation among domains, the Pearson correlation coefficient was investigated (Table 5). The results showed that there was a statistically significant correlation among all patient safety areas, with the Pearson coefficient exceeding 0.6 in all domains, indicating a high level of correlation within the tool.

## 4. Discussion

This study aimed to identify potential gaps in hospital readiness during the COVID-19 pandemic in Jordan by obtaining feedback from top, senior, and middle management within hospitals. Respondents were distributed between private and public hospitals across Jordan. The findings of this work revealed that the overall level of readiness, according to the six quality and patient safety domains, was acceptable (83.5%). The mean preparation of hospital emergency departments in response to disasters was assessed as 54.64% in a systematic review and meta-analysis reported by Kazemzadeh et al. (2019) to evaluate the preparedness of hospital emergency departments in reaction to disasters in Iran. The authors advised that hospital emergency departments consider planning and measures based on guidelines and accreditation criteria to improve disaster readiness [26]. Prior to the pandemic, Jordan’s healthcare system was already dealing with several issues, which were compounded by the COVID-19 catastrophe. However, there was little indication that decision-makers had properly used the pandemic’s early lockdown period to ensure Jordan’s healthcare system was fully prepared, particularly around the availability of Intensive Care Unit (ICU) beds, ventilators, and the health workforce. This had ramifications during the first pandemic wave that hit Jordan [27,28,29]. During the second and third waves of transmission, Jordan was in a grave health scenario due to the quick growth in daily infections and death toll, as well as high occupancy rates of isolation beds, ICU beds, and ventilators, as well as citizens’ adherence to COVID-19 precautionary measures being challenged [30]. Despite the efforts and effects of emergency preparedness activities, capacity building, and effective communication scored the least (2.46 and 2.48). During the pandemic, the healthcare system faced challenges due to the high pressure imposed by saturation of beds, loss, and absence of skilled/trained medical staff and high infection rates [6,28,31]. With this pandemic, the virus was novel, its epidemiology was changing in real time, and lessons were constantly being learnt. Therefore, skills and capacity were expected to be affected. Capacity building is critical in such situations to prevent, detect, investigate, and respond quickly to any changes and or updates in the management of COVID-19 cases [32]. Innovative approaches for virtual capacity building, training, and communication are required to distribute new perceptions, communicate best practices, and generate integrated communities of practice for all healthcare providers [32].

Hospitals in Jordan (*n* = 106) are mainly divided into private and public, with a total capacity of 12,081 hospital beds, where the public sector accounts for 67% of all hospital beds [28]. Our study further revealed statistically significant variations in preparedness according to sector and location. Apparently, private hospitals and those in the middle region demonstrated better performance. This could be attributed to the pressure on public healthcare facilities during the pandemic, given their limited funding and subsequent gaps in services. Similar results were reported in other countries. A study by Buzelli and Boyce [33] in Italy reported that the shortcoming of public hospitals’ preparedness during the pandemic was attributed to a decline in funding. Therefore, governments need to halt funding reductions to enhance performance in future emergencies [34,35]. Mature accredited hospitals (more than three accreditation cycles) in our study reported better preparedness during the pandemic when compared to immature facilities in two domains–emergency preparedness and infection control. Accreditation standards foster all elements and areas of emergency preparedness. However, the actual change takes time to be built into the organization and become a habit. Kennedy et al. (2022) investigated the results of preparedness-related accreditation that may impact emergency preparedness response and recovery capabilities in the United States. The research used data collected by the National Opinion Research Center based on a poll of health departments after a year of accreditation. It emphasized cross-cutting competencies that might aid in public health disaster preparedness, such as workforce development, quality improvement (QI) activities, evidence, and data-driven decision-making, as well as collaborations, accountability, and credibility. Results showed that with regards to workforce development, 89.3% of health departments emphasized that accreditation increased their ability to identify and solve training and capacity building issues. Accreditation also had a significant influence on health department QI initiatives, with 95.8% of respondents reporting that accreditation generated chances for quality and performance improvement inside their health department. Furthermore, the vast majority of respondents (92.5%) stated that information gleaned through QI and performance management activities impacts choices, resulting in more data-driven decision-making within the health department. Lastly, more than two third of respondents (68.7%) reported that accreditation also boosted the health department’s use of evidence-based methods. These findings provide evidence that, in addition to improving overall health department capacities, accreditation also supports the national guidelines, criteria, and activities identified as critical for developing and maintaining preparedness capability within the governmental public health system [36].

Furthermore, in research findings published in 2020 [37], the international society of infectious diseases addressed a request to maintain the focus on infectious illnesses, public health, and emergency response systems. They observed that unless the lessons acquired from COVID-19 are institutionalized, countries would be positioned to repeat COVID-19’s mistakes in the next pandemic.

### Strengths and Limitations

This is, to the best of our knowledge, the first research in the Middle East that investigated the role of accreditation as a quality and patient safety tool in emergency preparedness and upholding quality and safety during the COVID-19 pandemic. The use of patient safety domains that are based on a previously validated tool provided by the WHO provides strength to the study. Nevertheless, there are some limitations to the study. There is a limited number of studies that explored the effect of accreditation on emergency preparedness during COVID-19 pandemic worldwide and in the Middle East specifically, which reduced the ability to compare our findings with similar healthcare systems. The sample size of top, senior, and middle managers was not large due to the small population in this category in Jordan. However, a potential bias may occur during hospital selection and in-hospital distribution process. Furthermore, because the study design was based on quantitative methodology with pre-set responses, reflecting participants’ views and providing varied but useful qualitative information was not possible. The final limitation of the study is that due to the limited duration of the study fund and the lengthy process to obtain ethical approval, military hospitals were excluded from the study.

## 5. Conclusions

Although hospitals in Jordan demonstrated good preparedness during the COVID-19 pandemic, there are potential gaps when compared with accreditation standards. The key gaps were pertinent to capacity building and clear communication. Therefore, this study addresses those key areas that might benefit policy-makers, healthcare facility leaders, and accrediting organizations. There is an urgent need for novel capacity-building and communication strategies that reinforce behavioral changes towards better emergency preparedness. Furthermore, accrediting organizations’ standards that address capacity building and communication need to be shaped in a way that tackles new innovative teaching strategies that assess competency not only in normal situations but also during extreme emergencies.

## Figures and Tables

**Table 1 healthcare-11-00523-t001:** Demographic characteristics of participants.

Demographic Parameter	Frequency (%)
Educational level	
Diploma	26 (13.0)
Bachelor’s degree	127 (63.5)
Postgraduate MSc	34 (17.0
Postgraduate PhD	13 (6.5)
Gender	
Female	126 (63)
Age	
20–30 years	34 (17)
31–40 years	69 (34.5)
41–50 years	72 (36)
51 years and above	25 (12.5)
Managerial level	
Facility leaders	63 (31.5)
Senior management	76 (38)
Middle management	61 (30.5)
Position	
Hospital manager	10 (5)
Administrative director	26 (13)
Quality director	25 (12.5)
Infection prevention and control manager/coordinator	30 (15)
Safety manager	18 (9)
Nursing/medical manager	79 (39.5)
Others (training directors)	12 (6)

**Table 2 healthcare-11-00523-t002:** Participants’ hospital characteristics (the distribution was based on participants’ responses).

Parameter	Frequency (%)
Frequency of respondents according to their hospital type (*n* = 200)	
Private	104 (52)
Frequency of respondents according to their hospital location (region)	
North	83 (41.5)
Middle	98 (49)
South	19 (9.5)
Frequency of respondents according to their hospital’s accreditation status	
Not Accredited	15 (7.5)
Accredited (National Accreditation)	182 (91)
Accredited (International Accreditation)	3 (1.5)
Frequency of respondents according to their hospital’s accreditation cycles	
0–3	134 (67)
>3	64 (33)

**Table 3 healthcare-11-00523-t003:** Participants’ responses regarding the patient safety focus areas.

No.	Focus Area	Low (%) (1)	Moderate (%) (2)	High (%) (3)	Total Score (%)
1.	Emergency Preparedness				
1.1.	To what degree is the emergency response program helpful in dealing with COVID-19 pandemic?	2	42	56	84.7
1.2.	To what degree do the Bed Management Strategies contribute to increasing the availability of critical care beds in wards?	5.5	43	51.5	82
1.3.	To what degree is the Inventory Management System helpful in maintaining a sufficient number of supplies?	3	33.5	63.5	86.8
1.4.	To what degree do hospital standing committees (e.g., Governance and Leadership, Quality, and Environmental safety) contribute to advise/guide the management decisions during the COVID-19 pandemic?	6.5	33	60.5	84.6
2.	Infection prevention and control				
2.1.	To what extent does the provision of adequate supplies of personal protective equipment contributes to protecting employees from COVID-19?	2	26	72	90
2.2.	To what extent do the infection control and prevention program contribute to preventing the incidence of COVID-19?	4	36.5	59.5	85.2
2.3.	To what extent does the presence of guides and timetables for the proper cleaning and disinfection of surfaces reduce the occurrence of infection?	5	31	64	86.3
2.4.	To what extent are the surveillance activities (early monitoring and warning) helpful in preventing and controlling the incidence of COVID-19?	8.5	32.5	59	83.5
3.	Capacity building				
3.1.	To what extent are the educational materials regarding COVID-19 simplified and visible to the employees?	8	41	51	81
3.2.	To what extent does the continuous training and education program contribute to raising the efficiency of employees and increasing their awareness in dealing with the latest developments regarding COVID-19?	8	38.5	53.5	81.8
3.3.	To what extent does the competency assessment program help in improving employees’ knowledge and skills in dealing with the latest updates regarding COVID-19?	5.5	38.5	56	83.5
4.	Case Management				
4.1.	To what extent is the protection of patients’ rights, including access to care, convenient during the pandemic?	4.5	37	58.5	84.7
4.2.	To what degree does the triage system in the emergency department help in identifying any suspected cases and isolating them when identified?	5	33	62	85.7
4.3.	To what extent is the utilization of management protocols useful in dealing with COVID-19 cases?	4.5	32	63.5	86.3
4.4.	To what degree does the availability of referral/transport procedures aid in coordination with health authorities?	5	38	57	84
4.5.	To what degree does the documentation and recording system help in identifying and tracking the suspected cases?	6	38	56	83.3
5.	Communication				
5.1.	To what degree does the implementation of the information management system (communication plan) facilitate communication regarding the roles and responsibilities of managerial staff during this period?	4	40.5	55.5	83.8
5.2.	To what degree does sending (updated) key messages to staff, visitors, and the public, help in increasing awareness and disseminating accurate information?	4	40.5	55.5	83.83
5.3.	To what degree does the presence of a public information spokesperson enhance communication with the public, media, and supervisory health authority?	7.5	43.5	49	80.5
6.	Laboratory				
6.1.	To what degree does the availability of sufficient quantities of PPEs in the Laboratory help in protecting the lab staff from COVID-19?	4	28	68	88
6.2.	To what degree does the competency program help in increasing laboratory staff compliance with proper specimen collection?	3.5	37	59.5	85.3
6.3.	To what extent does the continuous training and education program contribute to raising the efficiency of nurses and laboratory technicians in the safe handling and transportation of specimens in accordance with national regulations?	4.5	35	60.5	85.3
6.4.	To what extent do the protocols and procedures of specimen collection and transportation contribute to preventing the incidence of COVID-19?	3	31	66	87.7

**Table 4 healthcare-11-00523-t004:** Stratification of participants’ response scores for the patient safety focus areas according to hospital type, location, accreditation status, and accreditation maturity level.

	Emergency Preparedness	Infection Control	Capacity Building	Case Management	Communication	Laboratory Service
Hospital Type	Score (from 3) Mean ± SD					
Public (*n* = 96)	2.38 ± 0.488	2.45 ± 0.541	2.34 0.577	2.47 ± 0.495	2.3819 ± 0.554	2.49 ± 0.532
Private (104)	2.68 ± 0.401	2.71 ± 0.386	2.58 ± 0.522	2.61 ± 0.476	2.57 ± 0.486	2.69 ± 0.426
Total (200)	2.54 ± 0.468	2.59 ± 0.483	2.46 ± 0.561	2.54 ± 0.489	2.48 ± 0.527	2.60 ± 0.488
*p* (ANOVA)	<0.001	<0.001	0.003	0.036	0.01	0.004
Hospital region						
North (*n* = 83)	2.37 ± 0.501	2.46 ± 0.537	2.31 ± 0.596	2.48 ± 0.513	2.36 ± 0.571	2.47 ± 0.546
Middle (*n* = 98)	2.68 ± 0.398	2.69 ± 0.403	2.57 ± 0.515	2.58 ± 0.478	2.55 ± 0.484	2.71 ± 0.419
South (*n* = 19)	2.55 ± 0.438	2.61 ± 0.509	2.58 ± 0.482	2.64 0.419	2.67 ± 0.444	2.57 ± 0.432
Total (*n* = 200)	2.54 ± 0.468	2.59 ± 0.483	2.46 ± 0.561	2.54 ± 0.489	2.48 ± 0.527	2.60 ± 0.488
*p* (ANOVA)	<0.001	0.005	0.003	0.236	0.016	0.003
Accreditation status						
Not Accredited (*n* = 15)	2.30 ± 0.392	2.40 ± 0.565	2.27 ± 0.715	2.39 ± 0.417	2.48 ± 0.530	2.48 ± 0.530
National (*n* = 182)	2.55 ± 0.471	2.6 ± 0.474	2.48 ± 0.547	2.55 ± 0.493	2.61 ± 0.485	2.61 ± 0.485
International (*n* = 3)	2.83 ± 0.289	2.67 ± 0.577	2.67 ± 0.578	2.67 ± 0.578	2.67 ± 0.578	2.67 ± 0.578
Total (*n* = 200)	2.54 ± 0.468	2.59 ± 0.483	2.46 ± 0.561	2.54 ± 0.489	2.48 ± 0.527	2.60 ± 0.488
*p* (ANOVA)	0.074	0.289	0.313	0.402	0.366	0.630
Accreditation maturity (number of accreditation cycles)						
Mature (>3) (*n* = 66)	2.59 ± 0.439	2.64 ± 0.423	2.48 ± 0.567	2.52 ± 0.513	2.52 ± 0.513	2.62 ± 0.467
Immature (≤3) (*n* = 134)	2.43 ± 0.511	2.49 ± 0.578	2.42 ± 0.55	2.4 ± 0.549	2.4 ± 0.549	2.56 ± 0.532
Total (*n* = 200)	2.54 ± 0.468	2.59 ± 0.483	2.46 ± 0.561	2.54 ± 0.489	2.48 ± 0.527	2.6 ± 0.488
*p* (ANOVA)	0.027	0.024	0.491	0.208	0.12	0.455

**Table 5 healthcare-11-00523-t005:** Correlation analysis for participants’ responses according to patient safety focus areas.

		Emergency Preparedness	Infection Control	Capacity Building	Case Management	Communication	Laboratory Services
Emergency Preparedness	Pearson Correlation	1	0.752 **	0.694 **	0.716 **	0.702 **	0.720 **
	Sig. (2-tailed)		<0.001	<0.001	<0.001	<0.001	<0.001
	*n*	200	200	200	200	200	200
Infection Control	Pearson Correlation	0.752 **	1	0.689 **	0.722 **	0.630 **	0.754 **
	Sig. (2-tailed)	<0.001		<0.001	<0.001	<0.001	<0.001
	*n*	200	200	200	200	200	200
Capacity Building	Pearson Correlation	0.694 **	0.689 **	1	0.702 **	0.686 **	0.687 **
	Sig. (2-tailed)	<0.001	<0.001		<0.001	<0.001	<0.001
	*n*	200	200	200	200	200	200
Case Management	Pearson Correlation	0.716 **	0.722 **	0.702 **	1	0.743 **	0.757 **
	Sig. (2-tailed)	<0.001	<0.001	<0.001		<0.001	<0.001
	*n*	200	200	200	200	200	200
Communication	Pearson Correlation	0.702 **	0.630 **	0.686 **	0.743 **	1	0.648 **
	Sig. (2-tailed)	<0.001	<0.001	<0.001	<0.001		<0.001
	*n*	200	200	200	200	200	200
Laboratory Services	Pearson Correlation	0.720 **	0.754 **	0.687 **	0.757 **	0.648 **	1
	Sig. (2-tailed)	<0.001	<0.001	<0.001	<0.001	<0.001	
	*n*	200	200	200	200	200	200

**. Correlation is significant at the 0.01 level (2-tailed).

## Data Availability

Not applicable.

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
