# Peer review of "Upholding Quality and Patient Safety during COVID-19 Pandemic—A Jordanian Case Study"

_healthcare, 2023, doi:10.3390/healthcare11040523_

Round 1

Reviewer 1 Report

The similarity rate in the article is higher than expected. Reconsideration is recommended.

Best regards

Author Response

Dear Esteemed Reviewer

Reviewer 2 Report

Does the introduction provide sufficient background and include all relevant references?

Principally, the introduction presents the most essential and indispensable research issues. The whole narrative is conducted in an accessible and correct way, mostly with respective and shrewd references in the footnotes. Albeit some debatable issues may be remarked.

One certain modification would have to be conducted. Namely, it was indicated that there is scarcity of conducted research to examine the relationship between accreditation standards and the COVID-19 response (see 111-112; similarly: 331-333). Meanwhile, not only a substantial number of such works have been published (eg. Heba Mohamed Adel, Ghada Aly Zeinhom, Raghda Abulsaoud Ahmed Younis, From university social-responsibility to social-innovation strategy for quality accreditation and sustainable competitive advantage during COVID-19 pandemic, Journal of Humanities and Applied Social Sciences 2021; 4(5): p. 410-437; D. Ferorelli, L. Nardelli, A. Dell’Erba, et. al., Medical Legal Aspects of Telemedicine in Italy: Application Fields, Professional Liability and Focus on Care Services During the COVID-19 Health Emergency, Journal of Primary Care & Community Health 2020;11, https://doi.org/10.1177/2150132720985; Cynthia A. Leaver, Joan M. Stanley,, Tener Goodwin Veenema, Impact of the COVID-19 Pandemic on the Future of Nursing Education, Academic Medicine 2022 Mar; 97(3): 82–89), but in addition they indirectly touched upon the research scope covered by the reviewed article (eg. Algunmeeyn A, El-Dahiyat F, Alfayoumi I, et al. Exploring staff perspectives of the barriers to the implementation of accreditation in Jordanian hospitals: Case studyInternational Journal of Healthcare Management 2021;14(4):1422–1428; Mohammed Hussein, Milena Pavlova, Wim Groot, An evaluation of the driving and restraining factors affecting the implementation of hospital accreditation standards: A force field analysis, International Journal of Healthcare Management 2022 (Published online: 11 Jun 2022): p.1-9, https://doi.org/10.1080/20479700.2022.2084810). Hence, this thesis (on scarcity of conducted research) does not seem fully convincing.

Further-mentioned change is recommended, however not indispensible. For it was noted that national and international organizations have developed various documents, which are usually linked with the national regulations and requirements (see 80), while in fact in the article's content there is no adequate justification of this thesis (e.g. indication of examples of such regulations or making respective references in a footnote). This thesis seems to be accurate, although the principles of proper methodology would decisively require a slightly broader argumentation here.

Are all the cited references relevant to the research?

The predominant part of references cited in the article have been prepared in an adequate, complete and correct manner. Regrettably some significant lapses have occured. Therefore some appropriate changes would be advisable.

To begin with, and what is the most relevant, there may be find a number of footnotes to the resources (webpages) that currently cannot be found: either removed or unavailable (see footnotes 9, 10, 19, 21, 23, 28, 30). Even worse, in the case of some, the date of access is not even given, which makes it even more troublesome to verify the accuracy of the references made in the footnotes (see footnotes 19 and 21).

Additionally, some of the footnotes are incomplete - the publication of the source was omitted (see footnotes 26, 35; for 35 it is probably: Glob Health Med. 2020 Apr 30;2(2)).

Moreover, it may be to note the lack of precise indication of exact page numbers, while it is possible - the source of reference cited in the footnote have the appropriate page numbering (see footnotes 1, 4, 5, 8, 15, 16, 20).

Is the research design appropriate?

The sequence of the presentation of the research acquis proposed in the reviewed article is accurate, as well as compatible with the classical scholar methodology applied to social sciences. All necessary elements were properly highlighted (introduction, materials and methods, detailed results, discussion, and conclusion).

Are the methods adequately described?

The research methods have been presented in detail and comprehensive manner, and also further references to the research sources used have been made. Particularly noteworthy is that results provides in tables present (in an immensely accessible way) very detailed data on the basis of which further research analysis was conducted.

Are the results clearly presented?

The results of the conducted research were properly presented. Nonetheless a slightly broader scope of the detailed features of participants might have seemed justified. This refers specifically to ethical subject. Namely, it have been mentioned, that study participants gave their informed consent (187-188). Whereas in the content of article there were no futher clarification what specific information were provided to participants. In view of the indicated lack of justification, it is problematic to determine whether (and in what scope) the consent have been given.

Are the conclusions supported by the results?

Principally the conclusions are properly justified and supported by the conducted research and its results. Worth mentioning is, that the article reasonable and accurate division to. Also It is worth mentioning that the article provides a reasonable and accurate division of the reaserch results respectively for private and public hospitals. Morover, significant strengthening as well as limitating factors of conducted study were correctly or satisfactorily indicated.

These above-mentioned remarks in turn determine the assesment on Originality / Novelty, Significance of Content, Quality of Presentation, Scientific Soundness, Interest to the readers.

In conclusion, the article may be accepted after minor revision.

Author Response

Dear Esteemed Reviewer

Reviewer 3 Report

The authors presented a cross-sectional study in Jordan to explore the correlation between compliance with accreditation standards and performance quality during the pandemic. The study used an online questionnaire and took place over three months between the 1st of March 2022 and the 30th of May 2022.

The introduction gives a reasonably detailed background regarding why the authors hypothesised that there was some correlation between the implementation of well-structured quality accreditation systems and hospitals' ability to respond to increased pressure and workload while maintaining a high standard of clinical care.

In the materials and methods section, the authors describe their inclusion criteria for the study. However, they did not include a detailed analysis of how many total hospitals were eligible, how many were excluded and for what reasons. This can be conveniently provided in a table or even as an appendix.

In the sampling strategy, the authors mention that 30 hospitals were involved --- how were they selected? (see my previous point regarding recruitment details)

hospital managers distributed the questionnaire within the hospital --- how did they select who to involve? Did they have any specific guidelines to follow?

In the strengths and limitations section, the authors correctly mention several limitations of the study but forget to list the potential selection bias in the selection and in-hospital distribution process. 

Author Response

Dear Esteemed Reviewer
